# Frequent Anti-V1V2 Responses Induced by HIV-DNA Followed by HIV-MVA with or without CN54rgp140/GLA-AF in Healthy African Volunteers

**DOI:** 10.3390/microorganisms8111722

**Published:** 2020-11-04

**Authors:** Frank Msafiri, Agricola Joachim, Kathrin Held, Yuka Nadai, Raquel Matavele Chissumba, Christof Geldmacher, Said Aboud, Wolfgang Stöhr, Edna Viegas, Arne Kroidl, Muhammad Bakari, Patricia J. Munseri, Britta Wahren, Eric Sandström, Merlin L. Robb, Sheena McCormack, Sarah Joseph, Ilesh Jani, Guido Ferrari, Mangala Rao, Gunnel Biberfeld, Eligius Lyamuya, Charlotta Nilsson

**Affiliations:** 1Department of Microbiology and Immunology, Muhimbili University of Health and Allied Sciences, Dar es Salaam P.O. Box 65001, Tanzania; agricolaj@yahoo.com (A.J.); aboudsaid@yahoo.com (S.A.); eligius_lyamuya@yahoo.com (E.L.); 2Division of Clinical Microbiology, Department of Laboratory Medicine, Karolinska Institutet, 17177 Stockholm, Sweden; charlotta.nilsson@ki.se; 3Division of Infectious Diseases and Tropical Medicine, University Hospital, LMU Munich, 80802 Munich, Germany; kathrin.held@med.uni-muenchen.de (K.H.); ynadai1@gmail.com (Y.N.); geldmacher@lrz.uni-muenchen.de (C.G.); akroidl@lrz.uni-muenchen.de (A.K.); 4German Center for Infection Research (DZIF), partner site Munich, 80802 Munich, Germany; 5Instituto Nacional de Saúde, Maputo 3943, Mozambique; raquelmatavele@gmail.com (R.M.C.); ednaviegas@gmail.com (E.V.); ilesh.jani@gmail.com (I.J.); 6MRC Clinical Trials Unit at UCL, London WC1V 6LJ, UK; w.stohr@ucl.ac.uk (W.S.); s.mccormack@ucl.ac.uk (S.M.); 7Department of Internal Medicine, Muhimbili University of Health and Allied Sciences, Dar es Salaam P.O. Box 65001, Tanzania; drbakari@yahoo.com (M.B.); pmunseri@yahoo.com (P.J.M.); 8Department of Microbiology, Tumor and Cell Biology, Karolinska Institutet, Nobel’s Rd 16, 17177 Stockholm, Sweden; britta.wahren@ki.se; 9Karolinska Institutet at Södersjukhuset, Södersjukhuset, 11883 Stockholm, Sweden; eric.g.sandstrom@gmail.com; 10The Henry M Jackson Foundation for the Advancement of Military Medicine, Bethesda, MD 20817, USA; mrobb@hivresearch.org; 11Imperial College London, London SW10 9NH, UK; s.joseph@imperial.ac.uk; 12Department of Surgery and Molecular Genetics and Microbiology, Duke University Medical Center, Durham, NC 27710, USA; gflmp@duke.edu; 13United States Military HIV Research Program, Walter Reed Army Institute of Research, Silver Spring, MD 20910, USA; mrao@hivresearch.org; 14Department of Global Public Health, Karolinska Institutet, 17177 Stockholm, Sweden; gunnel.biberfeld@ki.se; 15Department of Microbiology, Public Health Agency of Sweden, 17182 Solna, Sweden

**Keywords:** HIV vaccine, V1V2 antibodies, HIV-DNA/MVA, CN54rgp140 vaccine, African vaccinees

## Abstract

Antibody responses that correlated with reduced risk of HIV acquisition in the RV144 efficacy trial were assessed in healthy African volunteers who had been primed three times with HIV-DNA (subtype A, B, C) and then randomized into two groups; group 1 was boosted twice with HIV-MVA (CRF01_AE) and group 2 with the same HIV-MVA coadministered with subtype C envelope (Env) protein (CN54rgp140/GLA-AF). The fine specificity of plasma Env-specific antibody responses was mapped after the final vaccination using linear peptide microarray technology. Binding IgG antibodies to the V1V2 loop in CRF01_AE and subtype C Env and Env-specific IgA antibodies were determined using enzyme-linked immunosorbent assay. Functional antibody-dependent cellular cytotoxicity (ADCC)-mediating antibody responses were measured using luciferase assay. Mapping of linear epitopes within HIV-1 Env demonstrated strong targeting of the V1V2, V3, and the immunodominant region in gp41 in both groups, with additional recognition of two epitopes located in the C2 and C4 regions in group 2. A high frequency of V1V2-specific binding IgG antibody responses was detected to CRF01_AE (77%) and subtype C antigens (65%). In conclusion, coadministration of CN54rgp140/GLA-AF with HIV-MVA did not increase the frequency, breadth, or magnitude of anti-V1V2 responses or ADCC-mediating antibodies induced by boosting with HIV-MVA alone.

## 1. Introduction

Ultimate control of the HIV pandemic depends on availability of a vaccine that can curtail new HIV infections [1,2,3,4]. While many HIV-1 vaccine regimens have been tested in clinical phase I/II trials [5,6], only seven HIV-1 efficacy trials have been conducted to date [7,8,9,10,11,12,13,14]. In an ongoing HIV efficacy trial, an Adenovirus-26 vector prime and clade C gp140 boost regimen is being tested among women in sub-Saharan Africa [15], and among men who have sex with men and transgender people in the Americas and in Europe [16]. The ALVAC-HIV prime AIDSVAX B/E gp120 boost regimen given in the phase III RV144 efficacy trial in low-risk populations in Thailand, is so far the only HIV vaccine regimen to demonstrate moderate efficacy [12]. The pox prime–protein boost combination conferred a vaccine efficacy (VE) of 31.2% against HIV infection after 42 months [12], with an estimated VE of 59.9% at 12 months post initial vaccination reported in a subsequent post hoc analysis [17].

HIV-1 vaccine regimens consisting of recombinant DNA primes and vector-based boosts have been demonstrated to be safe and highly immunogenic [18,19,20,21,22]. The safety, immunogenicity, optimal dose, and route of delivery of a multigene, multiclade HIV-DNA priming vaccinations followed by boosting with heterologous HIV-1 modified vaccinia virus Ankara (MVA)-Chiang Mai double recombinant (CMDR) vaccine (HIV-MVA) have been evaluated in several phase I/II HIV vaccine trials [23,24,25,26,27,28]. Potent and durable immune responses were elicited after priming three times with HIV-1 DNA vaccine and boosting twice with HIV-MVA immunizations [29,30].

The TaMoVac II trial, a phase IIa HIV vaccine trial explored the safety and immunogenicity of a regimen with intradermal (ID) HIV-DNA priming with or without ID electroporation followed by boosting with modified vaccinia virus Ankara (HIV-MVA) given with or without an adjuvanted subtype C Env protein (CN54rgp140/GLA-AF) in healthy volunteers from Tanzania and Mozambique [31]. There were no safety concerns associated with the vaccines or their mode of delivery. ID electroporation increased DNA-induced IFN-γ ELISpot Gag response rates, but did not impact on Env-specific response frequency or magnitude. An overall 97% of vaccinees who completed all vaccinations had an IFN-γ ELISpot response to Gag or Env. The coadministration of CN54rgp140/GLA-AF significantly enhanced the magnitude of binding antibody responses, but not their overall frequency, which was very high at 95, 99, and 79% for subtype B gp160, C gp140 and E gp120 antigens, respectively. Neutralizing antibodies (NAb) were detected using the TZM-bl assay. NAb to CRF01_AE TH.023.06 were seen in both randomization groups. NAb responses to subtype C 93MW965 were significantly more frequent among vaccinees given HIV-MVA in combination with the adjuvanted protein boost. NAb to subtype B SF162 and to subtype C GS015.EC12 were rare, and none of the vaccinees generated NAb to CRF01_AE CM235.EC4. Overall, the titers of NAb responses were low in both vaccination groups [31].

The second variable loop (V2) of the HIV-1 envelope (Env) is important for HIV-1 infectivity, suggesting that immune responses to V2 may be important for vaccine efficacy [32,33]. The V2 loop participates in the binding of gp120 to CD4 receptors [34,35], and in the binding to the integrin α4β7 gut homing receptor [36]. Conserved residues in the V2 loop interact with the integrin α4β7 receptor on target cells leading to integrin LFA-1 activation and the formation of virological synapses [36].

Analysis of correlates of risk for HIV infection among HIV-infected and uninfected vaccine recipients in the RV144 trial revealed that vaccine-induced binding IgG antibodies to V1V2 Env protein as presented on murine leukemia virus gp70 scaffold inversely correlated with risk of infection [37]. Furthermore, high plasma levels of Env-specific IgA antibodies were directly correlated with risk of infection [37]. The RV144 vaccine regimen induced anti-V2 antibodies against conserved epitopes in the V2 loop of HIV-1 Env gp120 [38]. Among vaccinees with low levels of anti-Env IgA responses, IgG antibodies to linear epitopes in V2 and V3 loops were inversely correlated with infection risk [39]. Moreover, antibody-dependent cellular cytotoxicity (ADCC)-mediating antibodies were found to correlate with decreased susceptibility to HIV infection when low levels of anti-Env IgA responses were elicited [37].

IgG subclass has been shown to influence vaccine efficacy [40]. IgG1 and IgG3 have high affinity to Fc gamma receptor I (FcγRI) and demonstrate stronger antiviral properties than IgG2 and IgG4 [41,42,43]. In RV144, high levels of both IgG1 and IgG3 responses to the crown of V2 loop that mediated ADCC and antibody-dependent cellular phagocytosis (ADCP) activity were elicited in protected vaccinees [44]. However, protective immune responses waned in direct correlation with declining IgG3 levels, suggesting the functional relevance of V1V2-specific IgG3 responses in mediating protection against HIV-1 acquisition [44].

In the present study, we assessed antibody responses previously shown to be of importance for reduced risk of HIV acquisition in the RV144 trial. We describe responses to five dominant linear Env-epitopes, V1V2-specific total IgG, IgG1, and IgG3 responses, and anti-Env IgA antibody responses, as well as ADCC-mediating antibodies in TaMoVac II vaccinees.

## 2. Materials and Methods

### 2.1. Study Design

A multicenter, placebo-controlled, phase IIa HIV vaccine trial (TaMoVac II) was conducted in Tanzania and Mozambique, and safety and immunogenicity data have been reported previously [31]. Healthy volunteers were randomized twice in a factorial design; first to receive three doses of 600 μg HIV-DNA (HIV-1 gp160 subtypes A, B, C; Rev B; Gag A, B, and RTmut B) priming immunizations with or without electroporation (EP). The priming vaccinations were given using a needle-free ZetaJet™ device (Inovio Pharmaceuticals, Plymouth Meeting, PA, USA), at weeks 0, 4, and 12, either as two intradermal (ID) injections, or two ID injections with EP using the Derma Vax device (donated by Cellectis AS, Romainville, France), or one ID injection with EP. Thereafter, the participants were randomized into two groups; group 1 received two HIV-MVA (MVA-Chiang Mai Double Recombinant [CMDR], expressing Gag-Pol subtype A and a membrane-anchored functional HIV-1 gp150 Env subtype E) boosting vaccinations and group 2 received two boosts of HIV-MVA coadministered with an Env glycoprotein of HIV-1 subtype C, CN54rgp140. The boosting immunizations were delivered intramuscularly by syringe, 16 weeks apart. The glycoprotein was adjuvanted by glucopyranosyl lipid A in its aqueous form, GLA-AF (Table 1).

Here we report anti-V1V2 IgG and ADCC-mediating antibody responses using all available stored serum and plasma samples (*n* = 144) collected at baseline and four weeks after the last vaccination from vaccinees who completed all immunizations. The fine specificity of the antibody response to Env (linear epitopes) was mapped in a subset of 60 of these (28 vaccinees from group 1 and 32 vaccinees from group 2) and 11 placebo recipients. Samples for epitope mapping were selected by random assignment from all study sites. Analyses of anti-V1V2 IgG1 and IgG3 responses, and Env specific IgA antibodies were performed in samples collected from 57 vaccinees (32 from group 1 and 25 from group 2) recruited at the Muhimbili University of Health and Allied Sciences (MUHAS) HIV clinical trial site in Tanzania. We limited the testing of V1V2-specific IgG subclasses to the vaccinees at the MUHAS site due to the fact that there were no differences in total IgG V1V2 responses between the trial sites.

### 2.2. Linear Peptide Microarray

The fine specificity of antibody responses was mapped using a linear peptide microarray that was custom designed to include Env variants of the current global HIV pandemic [45], with Env sequences from eight recently transmitted HIV primary isolates belonging to subtypes A, B, C, CRF01_AE, and CRF02_AG. To further assess vaccine-specific IgG responses, HIV Env sequences of two HIV vaccines, HIV-MVA (CRF01_AE) and CN54rgp140 (subtype C), were added to the array. For fine mapping of previously identified hot spots of IgG recognition [39,45], up to 90 additional peptide variants covering these positions (V2 (HxB2 164–178), V3 (HxB2 300–324), V4 (HxB2 409–447), immunodominant region of gp41 (HxB2 576–614), and transmembrane cytoplasmic tail (HxB2 696–730) were included.

Peptide array mapping of linear Env-responses was conducted as described previously [45]. Briefly, after blocking the custom-designed peptide array slides (JPT, Berlin, Germany) plasma samples were diluted 1:100 and incubated for 2 h at room temperature (RT). Bound human IgG was detected using a Dylight649 labeled mouse antihuman IgG antibody (1:5000, 1 h at RT; JPT). Plasma from baseline and postvaccination visits of one participant was processed on the same day to avoid experimental bias. Array slides were scanned using a GenePix 4000A scanner at 650 (signal) and 532 nm (background) and images were analyzed using GenePix Pro 6.0 (Molecular Devices). After manual control of the array layout, results linking each peptide position with a fluorescence intensity (FI) value were exported into gpr files. Custom R-scripts were used to subtract baseline reactivity and to map the FI values of the corresponding peptides onto the assembled Env sequence of the 10 full-length Env sequences included in the array. FI values above 2500 after baseline value subtraction were considered positive. Mean FI values were then calculated over all the tested peptides at a given Env position if more than 25% of the participants showed a positive response.

Amino acid sequence logos were generated using WebLogo3 software [46] based on mean FI responses of all vaccinees against each peptide variant.

### 2.3. Assessment of Binding Antibodies to the V1V2 Region of HIV-1 Env

#### 2.3.1. Binding IgG Antibodies

Plasma samples collected at baseline and four weeks after the last vaccination were tested for vaccine-induced binding IgG antibodies to scaffolded gp70V1V2 protein of CRF01_AE (A244) and subtype C (CN54) using enzyme-linked immunosorbent assay (ELISA) as previously described [37] with minor modifications. U96 MaxiSorp Nunc-Immuno plates (Thermo Scientific, Roskilde, Denmark) were coated with 0.2 μg/well of recombinant gp70 (MLV)-V1V2 (HIV-1/AE/A244 or HIV-1/CN54, Immune Technology Corp, New York, NY, USA), and incubated overnight at 4 °C. Thereafter, 200 μL/well of blocking buffer was added and plates incubated overnight at 4 °C. Diluted plasma samples, titrated using two-fold dilutions, beginning at 1:100, were added in duplicates at 100 μL/well and the plates were then incubated at 37 °C for one hour. Protein–antibody complex was detected by adding 100 μL of sheep antihuman IgG conjugated with horseradish peroxidase (IgG-HRP, Binding Site, Birmingham, UK) to each well, and visualized using ABTS peroxidase substrate system (KPL, Gaithersburg, MD, USA). Optical density was read at 405 nm following incubation in darkness for one hour. A sample was deemed positive if the mean absorbance value after vaccination was more than twice that of the preimmunization sample. Results were reported as the reciprocal values of end-point titers.

#### 2.3.2. Binding IgG1 and IgG3 Antibodies

Vaccine-induced IgG1 and IgG3 antibodies to scaffolded gp70V1V2 protein of A244 were detected using ELISA as described above for total IgG antibodies using sheep antihuman IgG1-HRP or IgG3-HRP (Binding Site, Birmingham, UK).

### 2.4. Depletion of IgG from Serum Samples

For evaluation of vaccine-induced Env specific IgA antibodies, IgG was first depleted from serum samples. Depletion was performed using protein G Sepharose 4 Fast Flow beads (GE Healthcare, Buckinghamshire, UK) following manufacturer’s instructions with minor modifications. Protein G Sepharose beads were added to an Eppendorf tube, washed, and mixed with an equal volume of buffer to make 50% protein G Sepharose slurry. Thereafter, 40 μL of the protein G slurry was transferred to Eppendorf tubes, and 40 μL of serum sample diluted 1:1 in PBS was added, resulting in a serum dilution of 1:4. Eppendorf tubes containing a mixture of slurry and specimen were then incubated on an end-to-end rotator at 4 °C for 2 h, followed by centrifugation at 13,000 rpm for 1 min. Supernatants were collected and stored at −70 °C. Depletion of IgG reactivity was subsequently confirmed using HIV-1 subtype C CN54 gp140 antigen in an anti-Env IgG ELISA, as previously described [29].

### 2.5. Assessment of Env-Specific IgA Antibodies

Detection of IgA antibodies to subtype C (CN54) gp140 was performed using ELISA. U96 MaxiSorp Nunc-Immuno plates (Thermo Scientific, Roskilde, Denmark) were coated with 100 μL/well of HIV-1 subtype C gp140 (Polymun Scientific, Klosterneuburg, Austria) diluted to a final concentration of 0.5 µg/mL, and incubated overnight at 4 to 8 °C. Plates were then washed and blocked with 150 μL/well of buffer containing 20% fetal calf serum (Invitrogen) in PBS, for one hour at 37 °C. Thereafter, 100 μL/well of diluted IgG depleted serum samples starting at a dilution of 1:100 were added in duplicates and plates incubated overnight at 4 to 8 °C. Following washes, 100 μL/well of diluted biotinylated goat antihuman IgA (Southern Biotech, Birmingham, AL, USA) was added, and the reaction was visualized by the addition of horseradish peroxidase (HRP)-conjugated streptavidin (Southern Biotech, Birmingham, AL, USA) and OPD peroxidase substrate (SIGMAFAST OPD tablet set). The reaction was stopped by adding 50 µL of 3M sulfuric acid to each well and optical density was read at 492 and 620 nm. A result was considered positive if the mean absorbance value of post immunization sample was more than twice the preimmunization value. The results were reported as reciprocal end-point titers.

### 2.6. Assessment of ADCC-Mediating Antibodies

ADCC-mediating antibodies to HIV-1 Env were measured using a luciferase-based assay as previously described [27]. Peripheral blood mononuclear cells collected from healthy HIV-1 seronegative donors were used as a source of effector cells [47]. CEM.NKR._CCR5_ cells [48] infected with Env.IMC.LucR virus subtype CRF01_AE HIV-CM235- 2-LucR.T2A.ecto/293T(IMCCM235) (GenBank accession no. AF259954.1) was used as targets. A preparation of polyclonal purified IgG from HIV infected donors (HIVIG- obtained through the AIDS Reagent Program, Division of AIDS, NIAID, NIH) was used as positive control while serum from an HIV-uninfected individual was used as negative control. Biological activity of ADCC-mediating antibodies was measured as percentage reduction of luciferase activity in serum. The titer of ADCC-mediating antibodies was defined as the reciprocal of highest dilution showing more than 15% specific killing activity after background subtraction.

### 2.7. Statistical Analysis

Data analysis was performed using GraphPad PRISM version 7.0, R, and Windows Microsoft Excel. Fisher’s exact test was used to compare the frequency of V1V2 antibody responses between group 1 (HIV-MVA alone) and group 2 (HIV-MVA coadministered with CN54rgp140/GLA-AF). The Mann–Whitney test was used to compare the magnitude of antibody responses between the two groups and for comparing total IgG responses against linear Env epitopes. Wilcoxon matched–paired signed rank test was used to compare anti-Env CN54gp140 IgG reactivity in bulk and IgG-depleted serum samples. Observed differences in humoral immune responses were considered statistically significant at a *p*-value <0.05.

### 2.8. Ethical Considerations

The TaMoVac II (31) trial protocol was approved by ethical review boards of the Muhimbili University of Health and Allied Sciences (MUHAS), the Mbeya Medical Research Ethics Committee, and the National Institute for Medical Research (NIMR) in Tanzania, as well as the National Health Bioethics Committee in Mozambique, the Regional Ethics Committee in Stockholm, Sweden, and the Ethics Committee of the Ludwig Maximilian University in Munich, Germany. The Tanzania Food and Drugs Authority and the Pharmaceutical Department of the Ministry of Health in Mozambique approved use of vaccine candidates in Tanzania and Mozambique. Principles of Good Clinical Practice and requirements of International Conference on Harmonization (ICH-GCP) were adhered by all investigators. Participants were enrolled into the study after passing an assessment of understanding of informed consent.

## 3. Results

### 3.1. The Vaccination Regimen Induced IgG Antibodies Recognizing Linear Antigenic Epitopes in the V1V2, C2, V3, C4, and gp41 Env Regions

Mapping of IgG responses against the HIV-1 Env was conducted in plasma samples of 71 randomly selected trial participants four weeks after the last vaccination. Of these, 28 were from group 1; recipients of three HIV-DNA priming immunizations and two HIV-MVA vaccinations, and 32 were from group 2; recipients of three HIV-DNA priming immunizations and two HIV-MVA boosts coadministered with CN54rgp140/GLA-AF. In addition, 11 placebo recipients were included in the analysis.

Overall, the IgG responses in both boosting groups targeted similar Env regions (Figure 1, Appendix A). There were five immunodominant regions (IDR) as defined by a frequency of responders (FOR) of >60% (Table 2). In both boosting groups, IgG responses against the V2 (IDR1_V2; starting at amino acid positions 176; HXB2 aa164), V3 (IDR3; aa321–326; HXB2 aa300–305), as well as gp41 (IDR5_gp41; aa612; HXB2 aa580) Env regions were observed. For the V2 region, similar FOR were observed in both boosting groups; however, the mean fluorescence intensity (FI) was slightly higher in group 1 compared to group 2 (mean FI 20,587, range 0–43,536 vs. mean FI 14,486, range 0–58,227, *p* = 0.1854). The V3 region (HXB2 aa300–305), in contrast, was targeted in a higher percentage of vaccinees and responses were significantly stronger in group 2 (HXB2 aa304 mean FI 25,586, range 0–58,012 vs. mean FI 45,585, range 2734–60,770, *p* < 0.0001). IgG responses against the IDR5_gp41, covering part of the immunodominant region of gp41, were comparable in both groups. Two additional IDRs were observed in samples from group 2 (recipients of HIV-MVA plus CN54rgp140/GLA-AF): IDR2_C2 in the constant (C) region 2 (aa221; HXB2 aa200) and IDR4_C4 in C4 (aa461; HXB2 aa433), containing one of the amino acids that make up the CD4 binding site (aa438). IgG responses to IDRs 1, 3, and 5 detected in boosting group 1 and IgG responses against all 5 IDRs detected in boosting group 2 were significantly higher than in the placebo group (Figure 2).

Antigenic regions were considered immunodominant if recognized in >60% of participants of a boosting vaccination arm (HIV-MVA only *n* = 28; HIV-MVA + CN54rgp140/GLA-AF *n* = 32; Placebo *n* = 11). IDRs are listed with their respective position in the array, their corresponding HXB2 position, the Env region, and a representative amino acid sequence. The frequency of responders (FOR) and mean fluorescence intensity (FI) are stated for each position and vaccination arm. Background was subtracted as described in Material and Methods.

### 3.2. Comparison of Antigen Variant Recognition within the Five Immunodominant Regions

Additional peptide variants covering previously identified hot spots of IgG recognition included in the array allowed for fine mapping of the IgG response against four of the IDRs identified in the vaccinees. Preferred targeting of certain amino acid motifs was assessed in the context of the immunogen sequences (Figure 3).

Eighty-six peptide variants corresponding to IDR1_V2 (HXB2 aa163–177 instead of HXB2 aa164–176 due to differential cleavage of the peptides on the array) were included in the peptide array. Both boosting vaccination groups induced IgG targeting of the same 10 peptide variants. When comparing recognized variants (mean FI > 2500) with unrecognized variants, a preference of amino acids E^164^, K^169^, and VH^172–173^ was noted, which matched the HIV-MVA immunogen sequence (Figure 3A,E).

IDR2_C2 (HXB2 aa200–214), recognized by group 2 (recipients of HIV-MVA coadministered with CN54rgp140/GLA-AF), was only covered by two peptide variants (AITQACPKVTFDPIP and TMTQACPKVTFEPIP) in the peptide array. Of these, only AITQACPKVTFDPIP, matching the CN45 rpg140 protein sequence, gave a signal above background.

V3 tip amino acid sequence diversity of the HIV-1 Env was represented by 36 additional peptide variants (corresponding to HXB2 aa304–318) in the peptide array. We analyzed antibody recognition of these peptide variants in lieu of the whole IDR3_V3, spanning from HXB2 aa300–319 (Figure 3B,E). Of the 36 peptide variants corresponding to HXB2 aa304–318 present in the array, 26 were recognized by boosting group 1 (recipients of HIV-MVA), with four additional variants recognized by boosting group 2 (recipients of HIV-MVA coadministered with CN54rgp140/GLA-AF). These antibodies preferentially recognized amino acids S^306^, I^309^, and FY^315–316^. The recognized amino acid sequences closely matched all vaccine immunogen sequences, except for subtype B HIV-DNA.

IDR4_C4 (HXB2 aa433–447), only recognized by vaccinees in group 2 (recipients of HIV-MVA coadministered with CN54rgp140/GLA-AF), showed a relative conservation in the first seven amino acid positions, yet varied in the subsequent positions (Figure 3C/E). Of 41 IDR4_C4 variants present in the array, only five were recognized above threshold. The vaccinees’ antibodies showed a preference for hydrophilic amino acids at aa440 and 442 (K^44^° and Q^442^/N^442^), which matched closest to the CN54 rpg140 protein sequence.

The sequence homology within IDR5_gp41 (HXB2 aa580–595) peptides and immunogen sequences was relatively high, still, a preferred recognition of amino acids V^583^ and K/R^588^ in vaccine recipients was apparent (Figure 3D,E). Of 21 variants included in the array, only one was recognized by vaccinees in group 1, and 8 additional variants were recognized by vaccinees in group 2.

### 3.3. V1V2 Binding IgG Antibodies

Antibody responses to scaffolded gp70V1V2 antigen were determined in plasma from 144 vaccinees, four weeks after the last vaccination. High response rates of binding IgG antibodies to V1V2 protein of CRF01_AE and subtype C were induced in both boosting groups; 111/144 (77%) of vaccinees had binding IgG antibody responses to V1V2 of CRF01_AE and 93/144 (65%) to subtype C. No significant difference was found between the two vaccination groups. Anti-CRF01_AE V1V2 IgG responses were seen in 63/79 (80%) of recipients of HIV-MVA boost alone (group 1) and in 48/65 (74%) of participants boosted with HIV-MVA coadministered with CN54rgp140/GLA-AF, *p* = 0.4309. IgG antibodies to subtype C gp70V1V2 were induced in 52/79 (66%) of vaccinees in boosting group 1 and 41/65 (63%) of vaccinees in boosting group 2, *p* = 0.8611.

Similarly, the magnitude of anti-V1V2 IgG responses was not significantly different between the two boosting vaccination groups. Median titers of anti-CRF01_AE V1V2 responses were 800 (interquartile range (IQR); 200–1600) and 800 (IQR; 20–1600) among recipients of HIV-MVA alone and HIV-MVA + rgp140/GLA-AF vaccinees, respectively, *p* = 0.6754. Additionally, median titers of antisubtype C gp70V1V2 responses were 400 (IQR; 20–800) in HIV-MVA vaccinees and 400 (IQR; 20–1600) in HIV-MVA + rgp140/GLA-AF boosted participants, *p* = 0.7384 (Figure 4A,B).

### 3.4. Anti-V1V2 IgG1 and IgG3 Responses against CRF01_AE

To further define the IgG response, analysis of V1V2 specific IgG1 and IgG3 antibody responses was performed in a subset of individuals vaccinated at Muhimbili University of Health and Allied Sciences (MUHAS) in Dar es Salaam, Tanzania (*n* = 57). In this subset, no significant difference in frequency of responders or magnitude of total IgG antibody response to V1V2 region of CRF01_AE and subtype C was noted between the two boosting vaccination groups (Table 3, Figure 5A,B).

Anti-CRF01_AE V1V2 antibody responses were predominantly IgG1 in both boosting vaccination groups, 25/32 (78%) in recipients of HIV-MVA only and 19/25 (76%) among HIV-MVA + rgp140/GLA-AF vaccinees. However, anti-V1V2 IgG3 responses were significantly more frequent in vaccinees receiving HIV-MVA boosting alone, 12/32 (38%), than in participants boosted with HIV-MVA + rgp140/GLA-AF, 2/25 (8%), *p* = 0.0132 (Table 3).

There was no significant difference in the magnitude of anti-V1V2 IgG1 responses against CRF01_AE between the vaccination groups. Median titers in vaccinees boosted with HIV-MVA alone and in vaccinees receiving HIV-MVA coadministered with rgp140/GLA-AF were 800 (IQR; 400–3200) and 800 (IQR; 60–16,000), respectively, *p* = 0.1776 (Figure 5C). The level of V1V2-specific IgG3 responses to CRF01_AE is shown in Figure 5D.

### 3.5. Subtype C gp140-Specific IgA Binding Antibodies

IgG was depleted from serum samples of vaccine recipients in order to evaluate vaccine-induced Env-specific IgA antibodies. Env-specific CN54rgp140 IgG reactivity was significantly depleted as shown by an 86% reduction in optical density, from a median of 1.07 (IQR: 0.7–2.3) in bulk samples to 0.15 (IQR; 0.1–0.3) in IgG-depleted serum samples, *p* < 0.0001 (Figure 6A).

In total, 10/57 (18%) of vaccinees had IgA antibody responses to the Env protein. IgA antibodies were significantly more frequent in vaccinees from group 2 (HIV-MVA coadministered with adjuvanted CN54rgp140 boost) than in group 1 (HIV-MVA boost only), with a response frequency of 9/25 (36%) compared to 1/32 (3%) respectively, *p* = 0.0031. The level of IgA antibody responses to subtype C gp140 is shown in Figure 6B.

### 3.6. ADCC Activity against CM235 CRF01_AE Infected Target Cells

Serum samples from 135/144 vaccinees were available in sufficient volume for analysis. Overall, the frequency of vaccine-induced ADCC-mediating antibodies was low, detected in 29/135 (22%) of vaccine recipients. There was no significant difference in response rate between vaccinees who received HIV-MVA boosting alone (group 1), 18/75 (24%), and vaccinees boosted with HIV-MVA + CN54rgp140/GLA-AF (group 2), 11/60 (18%), *p* = 0.5282. Furthermore, no difference in magnitude of ADCC-mediating antibody responses to CM235 CRF01_AE infected cells was observed between the two arms of boosting immunizations. The median titer in responding vaccinees boosted with HIV-MVA alone was 1312 (IQR; 424–2496), while the median titer in the responding vaccinees boosted with HIV-MVA plus CN54rgp140/GLA-AF was 572 (IQR; 417–939), *p* = 0.125 (Figure 7).

## 4. Discussion

The role of an HIV vaccine in ending the global HIV/AIDS pandemic is indispensable [1,4,49]. Therefore, the search for an effective, affordable, and accessible HIV vaccine must go on [15]. In this study, we report induction of antibody responses that correlated with reduced risk of HIV acquisition in the RV144 efficacy trial [37,39]. In our study, vaccinees had received three HIV-DNA priming immunizations followed by two HIV-MVA boosting vaccinations given either alone or coadministered with subtype C CN54rgp140/GLA-AF [31].

Using a linear peptide microarray to assess linear HIV-1 Env B cell epitopes, IgG antibodies recognizing V2, V3, and gp41 immunodominant regions were detected after immunization with three HIV-DNA and two HIV-MVA with or without coadministration of CN54rgp140 GLA/AF. Responses against the C1 and C4 HIV-1 Env regions were only seen in vaccinees receiving HIV-MVA coadministered with subtype C CN54 rgp140/GLA-AF. A similar pattern, albeit with slightly varying frequencies and magnitudes in response, was previously reported for recipients of the same multivalent immunogens trialed consecutively, i.e., in vaccinees receiving three doses of HIV-DNA and two doses of HIV-MVA, followed by two immunizations of CN54 rgp140/GLA-AF [45]. Taken together, the two studies suggest that HIV-DNA priming and HIV-MVA boosting generates Env antibodies focused on V2, V3, and the gp41 immunodominant regions, while the addition of CN54 rgp140/GLA-AF boosting generates a stronger and broader V3 response, as well as recognition of additional Env regions, but does not enhance the V2 response.

When we analyzed IgG responses to linear V2 epitopes, reported to inversely correlate with HIV infection risk in the RV144 trial [37,39], we found that sequences of recognized V2 peptides were closely related to the HIV-MVA sequence and differed greatly from the CN54 protein sequence. This, together with the lack of any boosting of these responses after coadministration of CN54rgp140/GLA-AF, strongly suggests that HIV-MVA drives the anti-V2 IgG response. The somewhat lower antibody responses against V2 in vaccinees boosted with HIV-MVA and CN54rgp140/GLA-AF might suggest that the very strong V3 responses elicited by adjuvanted CN54rgp140 drive IgG responses away from V2. In light of our earlier findings, in which vaccinees receiving exclusively CN54 based immunogens (UKHVC 003SG) failed to elicit antibody responses against V2 [45], and in which the V2 response could be boosted by a single dose of HIV-MVA [50], we could speculate that the structural and sequence characteristics of the CMDR based MVA (HIV-MVA) immunogen used in this study, but not the CN54 based immunogens, elicited the anti-V2 IgG responses observed here. The suggested importance of the V2 sequences in the immunogen is further supported by the finding that responses against the same epitope located in the V2 loop, frequently targeted in vaccinees in the present study, were also detected in RV144 and Vax003 recipients for which the V2 epitope sequences of the immunogens (ALVAC-HIV and AIDSVAX B/E) match the HIV-MVA used here [38,39].

Of note, in all V2 peptide variants recognized by the vaccinees in the present study, a Lysine (K) was present at amino acid position 169. The sieve analysis performed following the RV144 trial indicated that vaccine-induced immune responses to Env V2 might have blocked viruses with a K at amino acid position 169 [51]. In addition, Gottardo et al., when studying IgG to linear Env epitopes to assess correlates of reduced risk of infection in RV144, reported a preferred targeting of peptides containing K^169^, with subtype B—which lacks K^169^—being the least targeted subtype [39].

The anti-V2 responses detected here in DNA primed and MVA boosted vaccinees seem to be of narrower breadth than reported in RV144 and Vax003 vaccinees [39], as only 10 out of the 86 tested HXB2 aa163 variants were recognized after vaccination. Such differences could either be inherent to the peptide array design and stem from the selection of peptide variants present on the arrays, or might be linked to the different immunogens. Nonetheless, we observe a similar preference of peptides present in Env sequences of subtype AE followed by subtype C. This is also in line with the more frequent IgG responses against scaffolded gp70V1V2 of subtype AE than C described herein.

In the current study, IgG antibodies binding to V1V2 Env of HIV-1 CRF01_AE were induced with high frequency in both vaccination groups, with an overall response rate of 77%. In the RV144 trial, in which participants were primed with a recombinant canarypox vector and boosted with two injections of AIDSVAX B/E Env gp120, 84% of vaccinees had binding IgG antibodies to the V1V2 loop of HIV-1 subtype E [38].

Here, nearly two-thirds of the vaccinees in both study arms exhibited binding IgG antibodies to subtype C gp70V1V2, with no significant difference in the magnitude of responses between the groups. Although the addition of the Env protein boost did not increase the frequency of responders or magnitude of IgG antibody responses to the subtype C CN54 V1V2 region, it was shown to significantly augment the magnitude of binding antibodies to homologous subtype C gp140 in the same vaccinees [31]. The increase in antisubtype C Env response is also in line with the results from the present peptide array analysis that showed an enhancement of IgG responses directed against the V3 (IDR3_V3) and the gp41 immunodominant region (IDR5_gp41), but not against the V2 region (IDR1_V2) in vaccinees given the CN54 protein together with the HIV-MVA boost.

IgG1 dominated the anti-V1V2 IgG subclass response, with three quarters of vaccinees developing binding IgG1 antibodies. In both immunization arms, most of the IgG1 and IgG3 reactivity was against V1V2 Env of HIV-1 CRF01_AE. In contrast to IgG3 antibodies that were detected more frequently in the HIV-MVA arm, there was no significant difference in proportion of responders or magnitude of anti-V1V2 IgG1 responses between the two groups. The dominance of V1V2-specific IgG1 responses reflects the concentration of each IgG subclass in the blood [52]. In addition, the RV144 vaccination regimen induced higher anti-gp70V1V2 IgG1 response rates than anti-gp70V1V2 IgG3 responses [53]. The importance of IgG3 antibodies was shown through the depletion of IgG3 antibodies from RV144 samples that resulted in a significant loss of ADCC and ADCP activity [44].

We also found that coadministration of Env protein with HIV-MVA boosting vaccination increased the induction of plasma IgA antibodies to homologous HIV-1 subtype C Env as compared to HIV-MVA boosting alone for which plasma IgA antibodies to subtype C Env were rare. High levels of vaccine elicited plasma IgA were found to have the potential to impact vaccine efficacy in the RV144 trial by potentially blocking the activity of ADCC-mediating antibodies binding to conformational C1 region of HIV-1 gp120 [54].

Here, the coadministration of adjuvanted CN54rgp140/GLA-AF with HIV-MVA did not enhance functional antibody responses. The frequency and magnitude of ADCC-mediating antibodies against CM235 CRF01_AE was low, with no difference between the two groups. The time interval between the two HIV-MVA vaccinations may have influenced the ADCC-mediating responses. In our previous phase I/II HIV vaccine trial, HIVIS03, in which the two boosting HIV-MVA injections were given 12 rather than 4 months apart as in the present study, high levels of ADCC-mediating antibodies against CM235 CRF01_AE were detected in 97% of vaccinees [27]. Karnasuta et al. made a similar observation when comparing antibody responses induced in RV144, VAX003, and VAX004 HIV vaccine efficacy trials. In the RV144 trial, the two protein vaccinations were given 12 weeks apart with ALVAC and after ALVAC priming, while in VAX003 and VAX004 the first two protein vaccinations were given within a month without vectored vaccine priming. The time interval between the first two protein immunizations is believed to have affected the breadth, maturity, and specificity of neutralizing antibodies elicited in the three efficacy trials [53]. Additionally, delayed boosting has been suggested as one method of improving HIV-1 vaccine efficacy [55]. Easterhoff et al. found that the breadth of V2-specific antibody effector functions was increased in a subset of RV305 clinical trial vaccinees who received a late AIDSVAX B/E boost [55].

In the present study, all vaccinees had received three priming immunizations with HIV-DNA encoding subtypes A, B, and C antigens prior to receiving two doses of HIV-MVA (CRF01_AE Env) with or without subtype C Env protein. CRF01_AE and subtype C V1V2-specific IgG responses were frequent and there was no clear difference between the boosting vaccination groups, suggesting little impact of the CN54gp140 subtype C protein boost and rather, arguing an important role for HIV-DNA. We report significantly stronger and broader anti-V3 responses in group 2 (CN54rgp140/GLA-AF coadministered with HIV-MVA) and these responses were accompanied by higher anti-Env IgA responses than were seen in group 1 (HIV-MVA-only). This would favor the HIV-DNA prime HIV-MVA boost vaccine strategy over HIV-DNA priming and CN54 rgp140/GLA-AF coadministration with HIV-MVA, since Gottardo et al. [39] found the presence of anti-V3 plasma IgG to be correlated with reduced infection risk only in RV144 participants with low anti-Env plasma IgA. While we found that the HIV-DNA/HIV-MVA vaccine concept may be favorable over the HIV-DNA/HIV-MVA/CN54rgp140/GLA-AF combination vaccine, we are still to understand the importance of the immune responses assessed here and cannot rule out additional immune responses as determinants of protective efficacy.

There are some limitations to this study. V1V2-specific IgG subclasses and Env-specific IgA were determined in a subset of individuals enrolled at the trial site at MUHAS, Dar es Salaam, Tanzania. We limited the number of tested subjects to the MUHAS site based on the fact that there were no differences in total IgG V1V2 responses between the trial sites. The subtype C CN54 gp140 antigen was selected for IgA antibody analysis based on the overall IgG antibody reactivity to the CN54 gp140 antigen in the TaMoVac II vaccinees. As reported previously, the frequency of IgG antibody responses was 99% to subtype C gp140, higher than that to subtype B gp160 (95%) and subtype E gp120 (79%) [31].

A phase IIb, HIV vaccine efficacy trial (PrEPVacc) is expected to start in East and Southern Africa in the last quarter of 2020 [56]. The trial will evaluate the effectiveness of combining pre-exposure prophylaxis with HIV vaccines in reducing HIV incidence. This three-arm, two-stage randomized trial will compare two experimental combination vaccine regimens with placebo control. Participants in one vaccination arm will receive four injections of DNA/AIDSVAX at weeks 0, 4, 24, and 48, while another group will be primed with a combination of DNA/CN54gp140 (weeks 0, 4) and boosted with MVA/CN54gp140 vaccines at weeks 24 and 48. Additionally, the same participants will be randomized to receive either daily TAF/FTC (Descovy) or daily TDF/FTC (Truvada) as pre-exposure prophylaxis in the first 26 weeks of immunization [56].

In summary, the HIV-DNA/MVA prime-boost regimen induced anti-V1V2 and anti-V3 responses, without corresponding induction of anti-Env IgA antibodies. Coadministration of CN54rgp140/GLA-AF with HIV-MVA induced stronger V3 responses but did not increase the frequency or magnitude of anti-V1V2 responses or ADCC-mediating antibodies, while increasing potentially inhibiting anti-Env plasma IgA responses. Taken together, these results support further development of the HIV-DNA/MVA prime–boost vaccine concept with or without protein boosting.

## Figures and Tables

**Figure 1 microorganisms-08-01722-f001:**
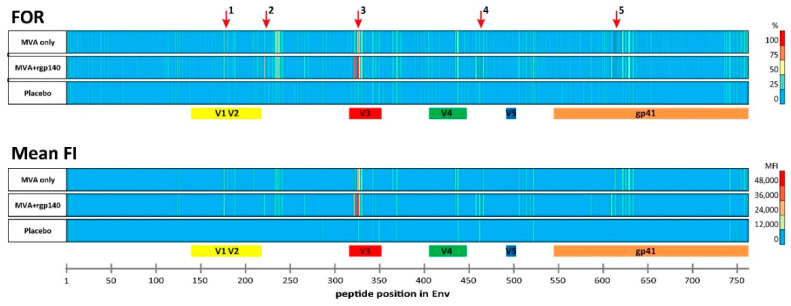
Heat map of frequency of responders (FOR) and mean fluorescence intensity (FI) plotted against individual antigenic regions along the entire HIV-1 Env as included in the peptide microarray. Heat maps of antigenic regions targeted by Env-specific IgG responses are shown for both vaccination groups and the placebo group four weeks after the last vaccination. Each row represents one of the boosting vaccination arms (HIV-MVA only *n* = 28; HIV-MVA + CN54rgp140/GLA-AF *n* = 32) and the placebo arm (*n* = 11). FI values corresponding to each peptide were mapped to the 10 full-length Env sequences included in the peptide array (HIV primary isolates subtypes A, B, C, CRF01_AE and CRF02_AG, and HIV vaccines HIV-MVA (CRF01_AE) and CN54rgp140 (subtype C)). Responses above 2500 FI after baseline subtraction were considered positive and the maximum FI was selected per position (Appendix A). The frequency of positive responses (>2500 FI) was calculated per peptide positions and is given as frequency of responders (FOR) in the upper graph. The mean FI depicted in the lower graph was calculated from the maximum FI per peptide position of each vaccinee per group and is shown for positive responses against peptide positions with a FOR >25%. Immunodominant regions (IDRs) 1–5 are indicated by red arrows and are listed in Table 2.

**Figure 2 microorganisms-08-01722-f002:**
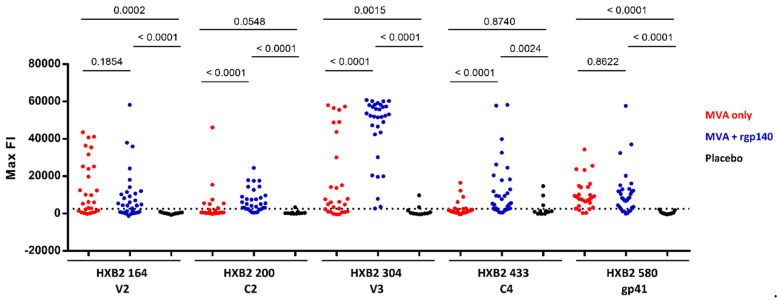
Statistical analysis of the IgG response against linear HIV-1 Env epitopes. Maximum FI of each vaccinee in the three vaccination arms given for the 5 IDRs. Maximum FI for each peptide position was chosen from the 10 full-length Env sequences included in the array. The dotted line represents the 2500 FI cut-off for positive responses. Corresponding HXB2 positions are given for all IDRs. The *p*-values comparing vaccination arms and placebo group were calculated using the Mann–Whitney U test. HIV-MVA only *n* = 28; HIV-MVA + CN54rgp140/GLA-AF *n* = 32; Placebo *n* = 11.

**Figure 3 microorganisms-08-01722-f003:**
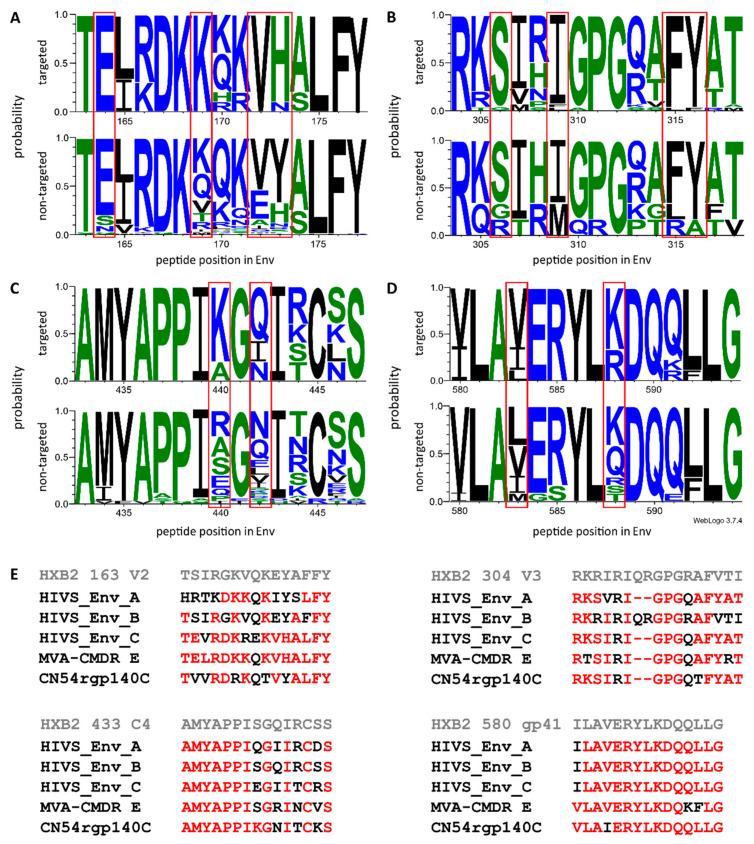
Preferred amino acid recognition at distinct Env positions in vaccinees. Sequence logos illustrating the probability for certain amino acids at given Env positions in peptide variants recognized (upper rows; mean FI >2500) and nonrecognized (lower rows; mean FI <2500) in vaccinees for four of the immunodominant regions (IDRs). Mean FI values were calculated per peptide variant across all vaccinees of each vaccination-boosting group. (**A**) IDR1_V2, HXB2 163; (**B**) IDR3_V3, HXB2 304; (**C**) IDR4_C4, HBX2 433; (**D**) IDR5_gp41, HXB2 580. The height of the letter depicts residue probabilities at a given amino acid position. Amino acids are colored according to their hydrophobicity (hydrophilic–blue, neutral–green, hydrophobic–black). (**E**) Vaccine and HXB2 sequences corresponding to these four IDRs. Amino acids with ≥0.6 probability of targeting the vaccine candidate sequences are highlighted in red. IDR2_C2 was only covered by 2 peptide variants in the array and is therefore omitted here.

**Figure 4 microorganisms-08-01722-f004:**
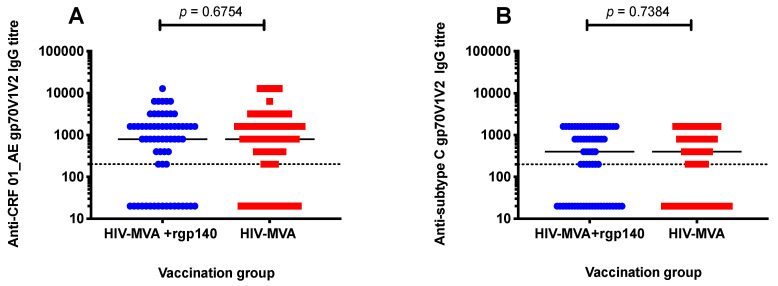
Vaccine-induced V1V2 IgG responses. Binding IgG responses to scaffolded gp70V1V2 protein of CRF01_AE (**A**) and subtype C (**B**) were determined in 144 vaccinees four weeks after the final immunization. Mann–Whitney test was used to compare the magnitude of anti-V1V2 IgG responses between the groups. No significant difference in the magnitude of binding IgG antibodies to V1V2 region of CRF01_AE and subtype C responses was observed between the two vaccination groups. The dotted line indicates the first dilution used. Results are reported as reciprocal values of end-point titers.

**Figure 5 microorganisms-08-01722-f005:**
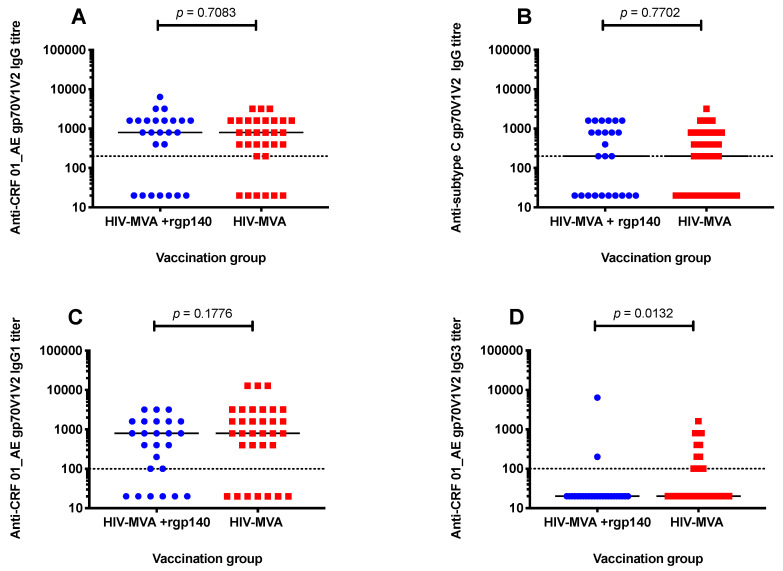
Binding total IgG antibodies to scaffolded gp70V1V2 protein of CRF01_AE and subtype C and IgG1 and IgG3 antibodies to scaffolded gp70V1V2 protein of CRF01_AE in plasma samples from a subset of vaccinees (*n* = 57). The Mann–Whitney test was used to compare the magnitude of V1V2 specific antibody responses between the two vaccination groups. There was no significant difference between the two vaccination groups in the magnitude of total IgG responses to CRF01_AE (**A**) or subtype C (**B**) V1V2 protein. No significant difference in the magnitude of anti-V1V2 IgG1 responses against CRF01_AE was seen between the two groups, *p* = 0.1776 (**C**). IgG3 antibodies binding to CRF01_AE V1V2 protein were significantly more frequent in recipients of HIV-MVA boost alone, *p* = 0.0132 (**D**). The dotted line indicates the first dilution used. Results are reported as reciprocal values of end-point titers.

**Figure 6 microorganisms-08-01722-f006:**
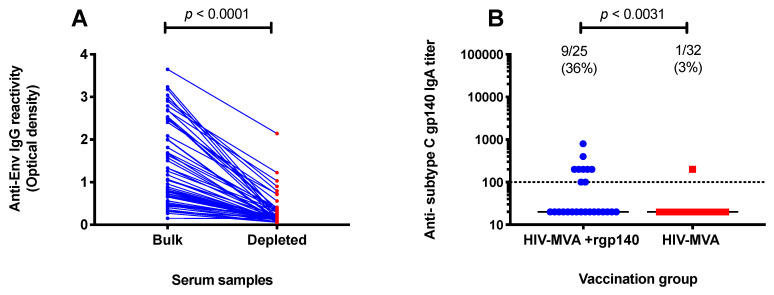
IgA antibodies to subtype C (CN54) gp140. IgG-depleted serum samples were used to assess induction of monomeric IgA binding to subtype C HIV-1 Env. Depletion of IgG reactivity was confirmed using anti-Env IgG ELISA (**A**). IgA antibodies were significantly more frequent in recipients of a combination of HIV-MVA and adjuvanted CN54rgp140 than in HIV-MVA vaccinees, *p* = 0.0031 (**B**). The dotted line indicates the first dilution used.

**Figure 7 microorganisms-08-01722-f007:**
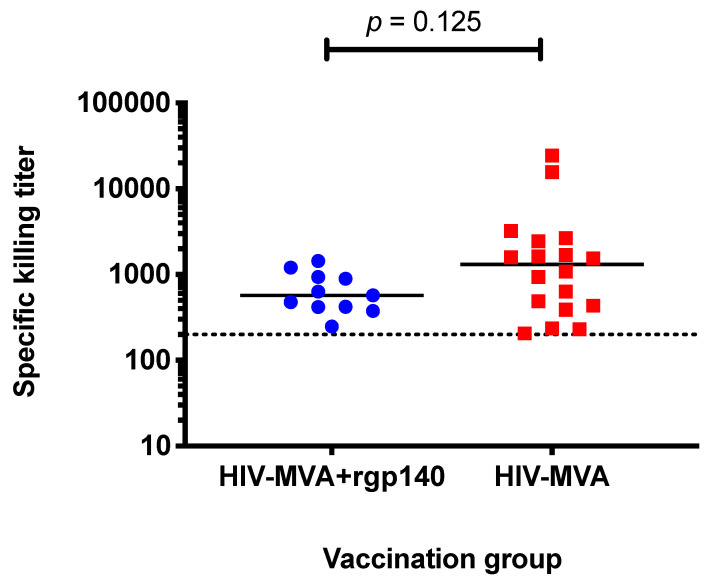
Magnitude of ADCC-mediating antibody responses in responding vaccinees. ADCC-mediating antibody responses to CM235 CRF01_AE infected cells were assessed using luciferase assay. No significant difference in titers of ADCC responses was seen among responding vaccinees in the two groups. Polyclonal purified IgG from HIV infected donors was used as positive control while serum from an HIV-uninfected individual was used as negative control. The titer of ADCC-mediating antibodies was defined as the reciprocal of highest dilution showing more than 15% specific killing activity (determined based on the responses seen prior to immunization to allow for 2% false positive rate) after background subtraction. The dotted line indicates the first dilution used.

**Table 1 microorganisms-08-01722-t001:** Overview of the vaccination schedule used in the TaMoVac II trial [31].

**First Randomization (Immunizations at Weeks 0, 4, 12)**				
**Group**	**A**	**B**		**C**
**Immunization ***	Two ID HIV-DNA injections (total 600 μg)	Two ID HIV-DNA injections (total 600 μg) with EP		One ID HIV-DNA injection (total 600 μg) with EP
**Second Randomization (Immunizations at Weeks 24, 40)**				
**Group**	**1**		**2**	
**Immunization**	10^8^ pfu HIV-MVA IM		HIV-MVA 10^8^ pfu IM plus 100 μg CN54rgp140/GLA-AF	

***** A total of 600 μg HIV-DNA was delivered per dose at each priming immunization (at week 0, 4, and 12).

**Table 2 microorganisms-08-01722-t002:** Summary of immunodominant regions detected in vaccinees.

IDR	Peptide Position	HXB2 Position	Env Region	Representative Sequence	FOR (%)			Mean FI		
					MVA Only	MVA+rgp140	Placebo	MVA Only	MVA+rgp140	Placebo
1_V2	176	164	V2	ELRDKKQKVHALFYK	68	63	0	20,587	14,486	0
2_C2	221	200	C2	AITQACPKVTFDPIP	25	72	9	0	9188	0
3_V3	321	300	V3	GNNTRKSIRIGPGQT	21	84	9	0	18,855	0
	322	301	V3	NNTRKSIRIGPGQTF	64	91	18	23,543	40,829	0
	325	304	V3	RKSIRIGPGSTFYAT	68	100	18	25,586	45,585	0
	326	305	V3	KSVRIGPGQTFYATG	82	97	27	28,859	43,633	12,656
4_C4	461	433	C4	AMYAPPIAGNITCKS	25	75	27	0	16,693	9640
5_gp41	612	580	gp41	VLAVERYLKDQKFLG	86	78	0	11,773	13,694	0

**Table 3 microorganisms-08-01722-t003:** Frequency of V1V2 antibody responses four weeks after the last vaccination.

Antibody	Antigen (gp70V1V2)	HIV Subtype	Frequency of Responses (Positive/Tested, %)		
			**HIV-MVA**	HIV-MVA + CN54rgp140/GLA-AF	*p* value
**IgG**	A244	CRF01_AE	25/31 (81)	18/25 (72)	0.5318
**IgG**	CN54	C	20/31 (65)	15/25 (60)	0.7859
**IgG1**	A244	CRF01_AE	25/32 (78)	19/25 (76)	>0.9999
**IgG3**	A244	CRF01_AE	12/32 (38)	2/25 (8)	**0.0132**

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
