# Peer review of "Frequent Anti-V1V2 Responses Induced by HIV-DNA Followed by HIV-MVA with or without CN54rgp140/GLA-AF in Healthy African Volunteers"

_microorganisms, 2020, doi:10.3390/microorganisms8111722_

Round 1
Reviewer 1 Report
Msafiri et al characterize Ab responses elicited in African volunteers receiving a DNA/MVA vaccine with or without gp140 in GLA boosting. Characterization includes mapping of Ab responses to linear peptide sequences, characterization of subgroup and isotype of elicited Ab and testing for ADCC using a luciferase assay.
The manuscript adds to the characterization of the Ab responses elicited by DNA/MVA+ protein vaccines. Several suggestions for the clarity of the data presentation are made.
- Lines 90-91: check sense. The test states that the gp140 enhanced nAb responses
- Table 1 is scrambled, clarify that 600 ug of DNA was delivered per dose
- Figure 1. Need to display data in another way. There seem to be a number of yellow lines outside of the 5 dominant epitopes, which are hard for me to discern. Could you display by graphing intensities with different color lines for +/- gp140 protein. Use median FI rather than mean FI.
- Figure 2 is more cogent than Figure 1. Could you leave Figure 1 out. You would need to define your 5 IDR in the text. That could perhaps be done in a revised Table 1. The mean FI should be given as medians of the means with ranges for each of the IDR. You could give the FI in thousands to let differences in the numbers stand out. A legend to Table 2 needs to give whether any background was subtracted. Column 1 could simply list the # of the IDR (give positions in the legend).
- Figure 5. There is no P value for panel D
- Fig 7, ADCC. It would be informative to also see the curves for ADCC activity for each individual tested. Maximum titers do not display the potential for prozones for non-neutralizing functions that require immune complexes to initiate protective innate responses.

Author Response
Responses to reviewer’s comments.
Dear Reviewer,
We would like to thank you for your valuable comments which have greatly improved our manuscript titled “Frequent anti-V1V2 Responses Induced by HIV-DNA Followed by HIV-MVA with or without CN54rgp140/GLA-AF in Healthy African Volunteers”.
We have addressed all the comments in the sections below and incorporated your suggestions into the revised version of the attached manuscript.
We hope that we have adequately addressed the comments raised and that with the changes made our manuscript is now suitable for publication.
Kind regards,
Frank Msafiri
Comments by reviewer 1:
- Lines 90-91: check sense. The test states that the gp140 enhanced nAb responses
Response to reviewer: We appreciate the comment. It is correct that neutralizing antibody responses to subtype C 93MW965 were significantly more frequent among recipients of HIV-MVA with adjuvanted protein boost than in vaccinees boosted by HIV-MVA alone. For clarity, we have added information on neutralizing antibody responses to other HIV-1 subtypes: subtype B SF162, subtype C GS015.EC12, CRF01_AE TH.023.06, CRF01_AE CM235.EC4 (page 2, lines 73-78).
- Table 1 is scrambled, clarify that 600 ug of DNA was delivered per dose.
Response to reviewer: We thank the reviewer for this observation. The table has been adjusted and a footnote added for further clarification. Additionally, for more information on the immunization schedule in the TaMoVac II trial, a reference to the study details is given in the table title.
- Figure 1. Need to display data in another way. There seem to be a number of yellow lines outside of the 5 dominant epitopes, which are hard for me to discern. Could you display by graphing intensities with different color lines for +/- gp140 protein. Use median FI rather than mean FI.
Response to reviewer: We appreciate the reviewer’s comments to Figure 1, and agree that the figure needed changing and more concise explanations. We, however, decided to maintain the graph as a heat map for several reasons. Firstly, when graphing the FI with different colour lines for the vaccination groups within one single graph, the individual lines will not be easily distinguishable due to significant overlap and information will be lost. If we were to show this data in line graphs, we would need to show 6 individual graphs which would overcrowd the figure. Secondly, putting the information of the IgG recognition of the entire HIV-1 Envelope Protein (Figure 1) solely into the text or as a table as suggested below, will lose all information of Env recognition outside of the 5 IDRs. We therefore decided to maintain the graph as a heat map, however, changed colouring from blue (no recognition) over yellow (infrequent or weak recognition) to red (frequent or strong recognition) to aid in the discernibility of the frequently and strongly recognised (red) from the infrequent or weakly recognised (yellow) peptides within the Env. Furthermore, we have added more concise explanations within the legend. The main goal of Figure 1 is to show that IgG recognition is targeted to few specific regions within the HIV-1 Env. A heat map is the most straightforward graph to achieve this and has been used for this purpose in previous publications (e.g. Gottardo, 2013 Fig 1 and 2).
We further maintain the use of the mean FI, as this is the established way of calculation for peptide binding values in studies using peptide micro arrays and therefore this will retain comparability between previously published micro array data (e.g. Stephenson, 2015; Joseph, 2017; Nadai, 2019; Joachim, 2020). We, however, always monitor the difference between mean and median FI during each analysis to avoid a mean FI biased through strong responses by one or few vaccinees. In order to depict the inter-individual variability of vaccine-induced IgG Env recognition we have added a supplemental Figure 1, which consist of a heat map depicting the max FI for each vaccinee. See results, section 3.1 (lines 273-289).
- Figure 2 is more cogent than Figure 1. Could you leave Figure 1 out. You would need to define your 5 IDR in the text. That could perhaps be done in a revised Table 1. The mean FI should be given as medians of the means with ranges for each of the IDR. You could give the FI in thousands to let differences in the numbers stand out. A legend to Table 2 needs to give whether any background was subtracted. Column 1 could simply list the # of the IDR (give positions in the legend).
Response to reviewer: Please refer in part to our answers above. We have decided to keep Figure 1, as it illustrates the vaccine-induced IgG reactivity against peptides covering all regions of the HIV-1 Env in a clear and straightforward manner. If we were to only retain Figure 2 and only define the IDR within the text, all information on Env recognition outside of the 5 IDRs would be lost.
Thank you for pointing out that we have omitted to state whether background was subtracted in the table legend. We have now added this information.
To better demonstrate the inter-individual variability, we have added a heat map depicting the max FI for each vaccinee to the supplement (Supplemental Figure 1). We consider it easier for the reader to acknowledge the variability between vaccinees, than to show the range in Table 2. The ranges also can be found in the text of the results, section 3.1 (page 6, lines 249-272).
- Figure 5. There is no P value for panel D
Response to reviewer: We thank the reviewer for pointing out this. The error has been corrected.
- Fig 7, ADCC. It would be informative to also see the curves for ADCC activity for each individual tested. Maximum titers do not display the potential for prozones for non-neutralizing functions that require immune complexes to initiate protective innate responses.
Response to reviewer: We thank the reviewer for this comment. When measuring ADCC-mediating antibodies, only CM235-infected cells were used as targets. Prozone effect is not seen when infected cells are used. The text suggesting that gp120-coated cells was used in the materials and methods section has been omitted as this was an error. The revised section is available on page 5, lines 209-219.

Reviewer 2 Report
In this manuscript, Msafiri et all perform a study looking at the responses of the TaMoVac II vaccinees. The study is a straightforward descriptive study measuring the responses to five dominant linear epitopes. The study is well done with standard methodology. Although the results is somewhat expected, it is still important for the vaccine community to determine if the CN54rgp140/GLA-AF adds any additional benefit. The sample size examined in the assays are adequate for the analyses performed. Comments: Abstract The authors should state a conclusion of the study in the abstract. Introduction: The authors should introduce the vaccine candidates being used in this study earlier in the introduction and it should be clear they are looking at correlates from the Rv144 trial. Methods How was the sample sizes determined? How was the subset of samples selected for different analyses? Was it informed by availability of samples or random assignment? The authors should clarify how this was done, and any potential biases should be noted. Line 123. The authors indicate 8 primary isolates but list 5 in the parenthesis. Results Fig 7. What was the control for the ADCC experiment? Was background subtraction done? If so it should be clear in the legend. Discussion No weakness or limitations of the study are noted in the discussion. Line 530 “These results support further development of 530 the HIV-DNA/MVA prime-boost vaccine concept.” I think it will benefit readers if the authors discuss this further. It seems addition of the CN54rgp140/GLA-AF adds no additional benefit. The implications of this should be discussed. Minor comment: It will be more readable if the authors abbreviate the CN54rgp140/GLA-AF earlier in the text.Author Response
Responses to reviewer’s comments.
Dear Reviewer,
We would like to thank you for your valuable comments which have greatly improved our manuscript titled “Frequent anti-V1V2 Responses Induced by HIV-DNA Followed by HIV-MVA with or without CN54rgp140/GLA-AF in Healthy African Volunteers”.
We have addressed all the comments in the sections below and incorporated your suggestions into the revised version of the attached manuscript.
We hope that we have adequately addressed the comments raised and that with the changes made our manuscript is now suitable for publication.
Kind regards,
Frank Msafiri
Response to reviewer 2
- Abstract: The authors should state a conclusion of the study in the abstract.
Response to reviewer: We thank the reviewer for this comment.
We have highlighted the conclusion that was written by adding “In conclusion, “ (lines 39-41).
- Introduction: The authors should introduce the vaccine candidates being used in this study earlier in the introduction and it should be clear they are looking at correlates from the Rv144 trial.
Response to reviewer: We agree with the reviewer and the introduction section has been revised. We have added a new paragraph about HIV-DNA/MVA vaccine trials (lines 56-62), and moved up the paragraph that describes the TaMoVac II trial to follow on the new HIV-DNA/MVA paragraph (lines 63-78). Furthermore, in the last paragraph of the introduction section, we have added a new sentence that we are looking for immune responses that were associated with reduced risk of acquiring HIV in the RV144 trial (line 101-102).
- Methods: How was the sample sizes determined? How was the subset of samples selected for different analyses? Was it informed by availability of samples or random assignment? The authors should clarify how this was done, and any potential biases should be noted.
Response to reviewer: We thank the reviewer for pointing out this. All available stored serum and plasma samples (n=144) were tested for anti-V1V2 IgG and ADCC-mediating antibody responses. Samples for epitope mapping (n=71) were selected by random assignment from all study sites. Assessment of anti-V1V2 IgG1 and IgG3 responses, and Env specific IgA was limited to the vaccinees (n=57) at the MUHAS site due to the fact that there were no differences in total IgG V1V2 responses between the trial sites. This information is now available in the Material and Methods section, Study design (lines 121-131).
- Line 123. The authors indicate 8 primary isolates but list 5 in the parenthesis.
Response to reviewer: We appreciate the reviewer’s comment.
There were 8 primary isolates used representing 5 subtypes. This has been clarified in the Materials and Methods section, line 138.
- Results Fig 7. What was the control for the ADCC experiment? Was background subtraction done? If so it should be clear in the legend.
Response to reviewer: We thank the reviewer for the comment. The figure 7 legend has been clarified to provide information on background responses. A preparation of polyclonal purified IgG from HIV infected donors (HIVIG- obtained through the AIDS Reagent Program, Division of AIDS, NIAID, NIH) was used as positive control while serum from an HIV-uninfected individual was used as negative control. The titer of ADCC-mediating antibodies was calculated after subtraction of the background. See Materials and Methods (lines 209-219), and legends under figure 7 (lines 425-433).
- Discussion: No weakness or limitations of the study are noted in the discussion
Response to reviewer: We thank the reviewer for the comment. Limitations are present in the manuscript, however, for clarity, a new sentence has been added in line 544.
- Line 530 “These results support further development of 530 the HIV-DNA/MVA prime-boost vaccine concept.” I think it will benefit readers if the authors discuss this further. It seems addition of the CN54rgp140/GLA-AF adds no additional benefit. The implications of this should be discussed.
Response to reviewer: The implications of our results have been further discussed.
The following has been added (lines 540-543):
“While we found that the HIV-DNA/HIV-MVA vaccine concept may be favorable over the HIV-DNA/HIV-MVA/CN54rgp140/GLA-AF combination vaccine, we are still to understand the importance of the immune responses assessed here and cannot rule out additional immune responses as determinants of protective efficacy.”
Furthermore, we have maintained the last sentence (line 565-566) and added “with or without protein boosting”.
